# Skeletal muscle cells opto-stimulation by intramembrane molecular transducers

Ilaria Venturino[1,2,4], Vito Vurro[2,4], Silvio Bonfadini[2], Matteo Moschetta[2], Sara Perotto [2], Valentina Sesti[2,3], Luigino Criante[2], Chiara Bertarelli[2,3] & Guglielmo Lanzani [1,2✉]

Optical stimulation and control of muscle cell contraction opens up a number of interesting applications in hybrid robotic and medicine. Here we show that recently designed molecular phototransducer can be used to stimulate C2C12 skeletal muscle cells, properly grown to exhibit collective behaviour. C2C12 is a skeletal muscle cell line that does not require animal sacrifice Furthermore, it is an ideal cell model for evaluating the phototransducer pacing ability due to its negligible spontaneous activity. We study the stimulation process and analyse the distribution of responses in multinuclear cells, in particular looking at the consistency between stimulus and contraction. Contractions are detected by using an imaging software for object recognition. We find a deterministic response to light stimuli, yet with a certain distribution of erratic behaviour that is quantified and correlated to light intensity or stimulation frequency. Finally, we compare our optical stimulation with electrical stimulation showing advantages of the optical approach, like the reduced cell stress.

[1] Dipartimento di Fisica, Politecnico di Milano, Milano, Italy. [2] Center for Nano Science and Technology, Istituto Italiano di Tecnologia, Milano, Italy. [3] Dipartimento di Chimica, Materiali e Ingegneria Chimica "Giulio Natta" Politecnico di Milano, Milano, Italy. [4] These authors contributed equally: Ilaria Venturino, Vito Vurro. ✉email: guglielmo.lanzani@iit.it

Contractile cells are a fascinating class of excitable cells. Due to their contraction ability, they are responsible for a massive amount of macroscopic actuation processes occurring in the human body. Muscle cells can work as pumps (heart), actuators (skeletal muscle), light collectors (in the iris) and fluid displacement controls (gastro enteric muscle). They are characterized by a high power-to-weight ratio, robustness, self-repair ability and efficiency[1]. All these properties candidate muscle cells and tissues as an intriguing active element when integrated with an elastic substrate, realizing a bio-hybrid system. These devices are achieved by harnessing different biological elements such as flagella[2], insect larvae[3], cardiac cells[4] or skeletal muscle cells[5]. They may lead to disruptive innovation in fields like medicine[6], environmental monitoring[7], sensing[8] and robotics[9–11]. The field, albeit promising and potentially disruptive, is still in its infancy and several issues remain open. One big issue regards the ability to control the cell's contraction without affecting its viability or functionality. Muscle-based bio-hybrid systems have been triggered mainly electrically by placing a pair of electrodes nearby it. The electric triggering is a golden standard in the field due to its simplicity, easiness of handle and control.

However, this approach has limits and drawbacks. One is low spatial resolution and low selectivity, due to spreading of the electric field that cannot be confined in a specific small area. This hampers the independent activation of different muscle fibres within the same sample. Currents may induce an inflammatory response or the generation of reactive oxygen species[12] leading to tissue damage and reducing its lifetime. Furthermore, wirings and connections can be cumbersome. For this reason, scientists are looking for other triggering mechanisms of muscle cells contraction and light has emerged as an appealing candidate due to its high spatial and temporal resolution. Light stimulation gets rid of wiring and it is in general less invasive or harmful, while providing high space and time selectivity. Cells, however, including muscles, are not light sensitive therefore specific strategies are in demand to use light as a stimulating tool[13]. Direct light stimulation[14] of the cytoplasm using infrared radiation, organic and inorganic semiconductor interfaces[15–18], and optogenetics[19–22] has been successfully exploited for living cell photo-stimulation.

In this work, we propose to use an amphiphilic amino-azobenzene derivative, named Ziapin2[23,24] as a light-driven triggering element for skeletal muscle cells. Molecular dynamics simulation has been used to demonstrate the phototransducer ability to dwell in plasma membrane due to simple electrostatic interaction[25]. In membrane environment, photoexcitation causes Ziapin2 isomerization leading to a membrane thickness modulation and a subsequent capacitance variation. This capacitance change has been demonstrated to induce a membrane potential modulation[26] characterized by a hyperpolarization, followed by a transient depolarization. Interestingly this process has been observed in several cell models[25,27,28] and furthermore, it has been demonstrated to trigger the generation of action potential in excitable cells and tissue[29].

Here, for the first time, we apply Ziapin2 to the C2C12 cell line. C2C12 myotubes are the ideal component for engineering bio-hybrid machines for several reasons. They are obtained from a commercially available cell line, and they are fairly easy to be cultured. C2C12 are skeletal muscle cells that exhibit negligible spontaneous activity with respect to cardiomyocytes. In principle, this allows for full control of their contraction, while in cardiomyocyte, the light stimulation is in competition with the spontaneous activity. In addition, being a cell line, C2C12 does not imply animal sacrifices. This cell model needs to be differentiated from myoblast into myotubes in order to show contractile ability. This process takes place by the single cells fusion into multinucleated myotubes. Cell alignment as well as substrate stiffness can be used to achieve this goal. Anisotropic alignment is quite efficient and can be obtained by creating a surface morphology[30] (surface pattern[31,32], nanofibers[33–35] and wrinkles[36,37]) highlighting a wide range of efficient pattern dimensionality[38]. Similarly, results can be obtained by printing extracellular adhesion proteins on the substrate[39,40].

We successfully set up a growth technique that allows cell maturation into multinuclear tubules by exploiting adhesion-patterned protein and we demonstrate a Ziapin2-mediated light stimulation of muscular tissue contraction. We report a characterization of the beating and contraction behaviour, as well as the performance in terms of control, energy consumption and lifetime. Finally, we compare our approach with that one based on electrical stimulation. Essentially the two methods yield similar results, even if the primary stimulation mechanism is different. However, it appears that at high workload optical stimulation is less harmful and less energy demanding then the electrical one.

## Results

**Myotubes formation and anisotropic tissue development**. We fabricated the patterned substrates by the micro-contact-printing (MCP) technique[41]. Briefly, MCP exploits a patterned stamp, usually made of polydimethylsiloxane (PDMS), to transfer adhesion proteins onto a substrate following the stamp pattern[42]. While PDMS stamps are usually fabricated using classical photolithographic techniques, we have taken advantage of a direct and maskless writing technique to imprint a user-customizable stamp surface by ultra-short pulse laser ablation fs-micromachine. In this way, the necessary production steps are simplified and drastically reduced. SEM images of the realized stamps are reported in Fig. 1a. We used the stamps to transfer the pattern over a cured PDMS film processed *via* spin coating. An emissive molecule, rhodamine B, was used to test the stamp transfer efficiency (Supplementary Fig. S1.a.). After that, we printed an extracellular matrix (fibronectin) over the PDMS films, and we seeded the C2C12 cells on top[43,44]. As shown in Fig. 1b, the cells seeded onto the patterned substrate are well aligned, at variance with cells grown onto the untreated substrate. More quantitatively, free cell growth leads to a broad distribution with an average orientation of 101.5 ± 52.9, while the aligned cells are confined into a much narrower distribution with the average orientation of 22.3 ±22.2 with respect to the direction of the pattern, as reported in Fig. 1c. The stamps are designed to have a line width of 84.7 ± 4.0 μm and lines distance of 27.8 ± 2.4 μm. These parameters are suitable for myotubes formation and growth[42,45]. The formation of the myotubes requires several days. First, the cells need to reach confluence after that they start to differentiate. We tested cells ten days after the confluence in accordance with literature reporting that after 7-10 days, the myotubes are mature enough to contract[46–50]. We assess the differentiation degree of the myotubes though the fusion index and the myotubes length[46,49]. The fusion index is determined by the percentage of nuclei inside myotubes, cells with at least two nuclei, over the total number of nuclei. Our experiments are characterised by a fusion index of 22.2 ± 1.0 % and a myotubes length of 321.4 ±89.1 μm, a value comparable with the literature[51,52]. Supporting Figure S1.b reports an example of the image used to perform this analysis. Figure 1d shows the timeline used for the cell maturation and the medium that we applied to improve the efficiency of the process.

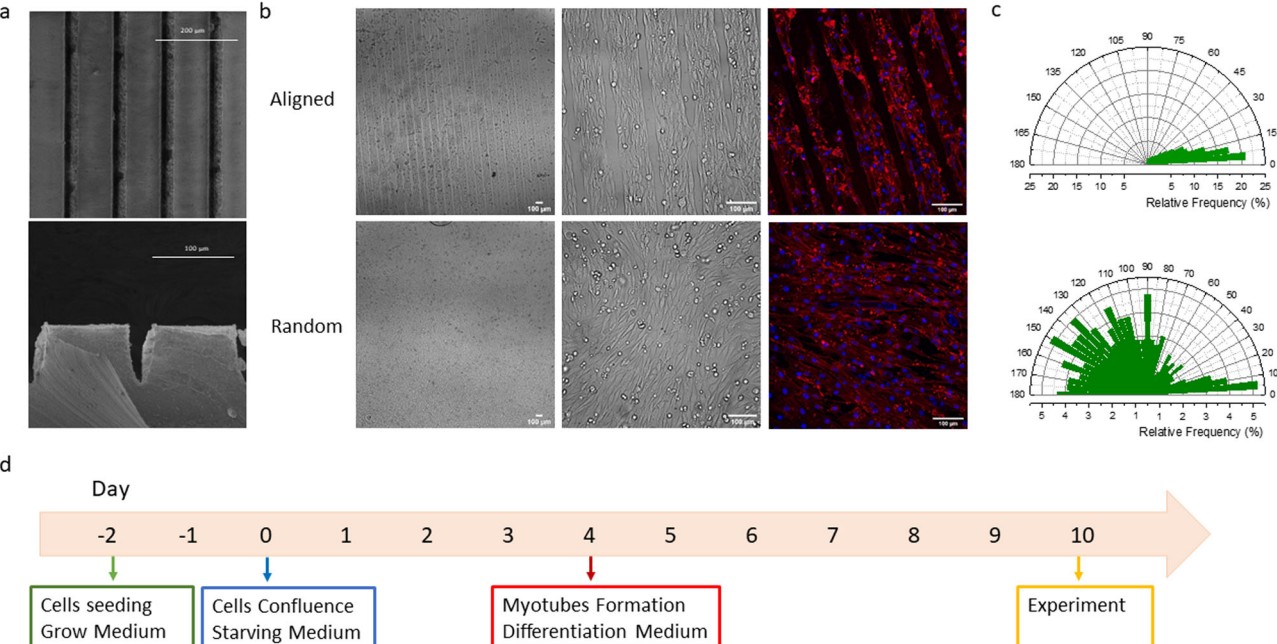

**Fig. 1 Cell growth. a** SEM images of the PDMS stamp. Front view and cross-section. **b** Bright field and fluorescent images represent: first raw aligned cells and second raw random arranged cells. The cells in the fluorescent images are stained with Hoechest (blue) and CellMask Plasma Membrane (red). **c** Nuclear distribution of aligned and random cells. (Aligned cells $f = 10$ and $r = 1$, random cells $f = 3$ and $r = 1$ **d** Representation of the medium changes necessary to induce muscle cell differentiation and the time line.

**Effect of Ziapin2 on the cells' membrane**. Once the differentiation step was completed, we started investigating the photopacing effect of Ziapin2. First, we added a diluted aqueous solution of Ziapin2 to the cell medium to a final concentration of 25 µM (see "Methods" for details). As previously described Ziapin2 dwells into the membrane of the cells inducing light sensitivity. Ziapin2 photoisomerization induces a cell membrane hyperpolarization, followed by a rebounded depolarization. This membrane potential modulation is produced by a membrane thickness variation causing a change in capacitance. This mechanism has been demonstrated in several cell models suggesting its independence by specific cell line. In particular Ziapin2-mediate photostimulation process in hIPSC-CMs has been used to studied Ziapin2 ability to start the excitation-contraction coupling. Aiming to have also an experimental confirmation about the photostimulation mechanism, we check the initial process (hyper/depolarization) by patch clamp technique. After the internalization of the photochromic molecule, which occurs in 10 min, light stimulation $(\lambda_{ex} = 470\,\text{nm})$ is applied to the cells through an inverted microscope delivering a power density of 50 mW mm$^{-2}$ and light pulses of $200 \pm 0.1$ ms length. The results, reported in Fig. 2, are in good agreement with the literature.

**Light-driven-cell contraction**. To induce cells' contraction, they were stimulated by 30 s pulse trains with different pulse frequencies, namely 0.5 (Supplementary Video 1), 1 (Supplementary Video 2) and 2 Hz (Supplementary Video 3). To visualize an area with a relevant number of myotubes we used a ×20 objective, and a power density of 42 mW mm$^{-2}$. We evaluated the cell mean contraction frequency, the optical triggering efficiency and the cell response statistics. Light-induced cell contraction was evaluated by analysing videos of the sample under illumination using a Matlab algorithm based on a visual recognition code. The algorithm tracks and detects the cell's contraction by selecting a

region of interests (ROI) and following it through each frame of the video (Fig. 3a). It is thus possible to obtain the position of the myotubes in each frame and to reconstruct the contraction profile of the ROI, as reported in Fig. 3b. By measuring the time between two consecutive contractions, we obtain the beating period and, consequently, the mean contraction frequency of the analysed cell. We collected all the contraction frequencies of the cells stimulated at a specific contraction frequency and we evaluated the mean value of the analysed population. We compare the mean contraction frequency of the population with the stimulation frequency: the closer these two values are the better this population is synchronized to the stimulation. While the contraction behaviour is clearly deterministic, we notice that myotubes are better synchronized at lower frequency. Indeed, when we used a stimulation frequency of 0.5 Hz, the mean contraction frequency was $0.48 \pm 0.04$ Hz and the histogram in Fig. 3c highlights that just a tiny fraction of the cells shows a contraction frequency slower than the stimulation one. At 1 Hz stimulation frequency, the mean contraction frequency was $0.93 \pm 0.13$ Hz. Again, Fig. 3c shows that the majority of the myotubes (70%) are characterised by a contraction frequency of 1 Hz, but the number of outliers increases. When the stimulation frequency is 2 Hz, only 40% of the analysed cells can reached the target frequency, and the measured mean contraction frequency was $1.44 \pm 0.32$ Hz. The misbehaviour of the cells at 2 Hz, that is a frequency that in principle can be well supported by skeletal muscle cells, can be related to a shortage in cell differentiation or maturation, or perhaps a consequence of being at room temperature. (There are possible actions to improve maturation[53–56] but this is out of scope here, so we focused on the optical triggering mechanism). At 2 Hz stimulation, the average frequency reported is 70% of the value. This implies that there are number of failures, or delays following the light pulse. Nevertheless, there are sequences of consistent contraction events, and those occur at 2 Hz.

To compare optical stimulation with electrical stimulation we inserted in a well a pair of platinum electrode, both placed inside

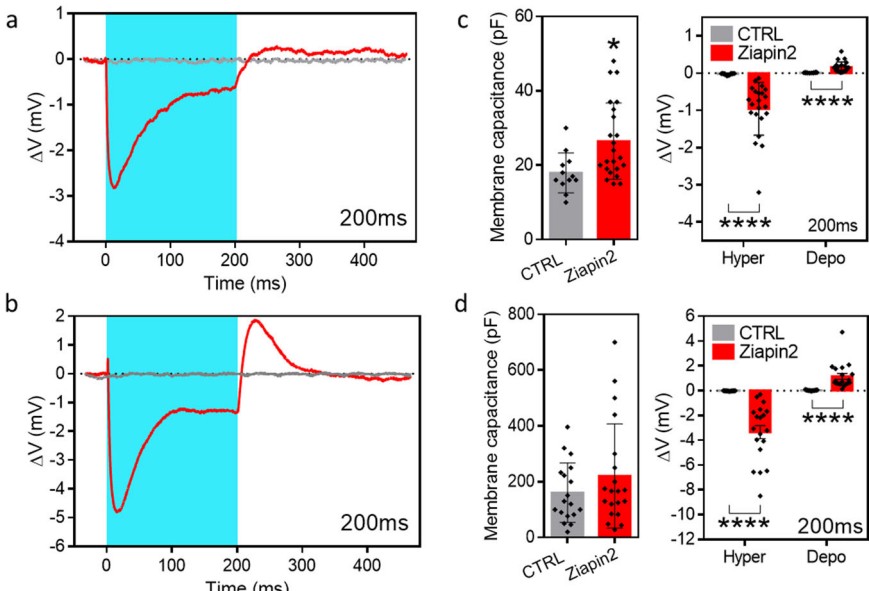

**Fig. 2 Patch clamp. a** Representative whole-cell current clamp traces recorded from C2C12 myoblasts either in the absence (CTRL, grey) or loaded with 25 µM of Ziapin2 (red) illuminated at 50 mW/mm² for 200 ms. **b** Representative whole-cell current clamp traces recorded from C2C12 myotubes either in the absence (CTRL, grey) or loaded with 25 µM of Ziapin2 (red) illuminated at 50 mW/mm² for 200 ms. **c** Plots representing membrane capacitance (left), peak depolarization and peak hyperpolarization in C2C12 myoblasts subjected to 200 ms (right) of light stimulation in the absence (CTRL, grey) or loaded with 25 µM of Ziapin2 (red). Black dots represent the single values. Data are expressed as mean ± sd. Unpaired Student's *t* test/Mann–Whitney *U* test; *$p < 0.05$, ****$p < 0.0001$; $n = 12$ and 22–23 for CTRL and Ziapin2, respectively. (**d**) Plots representing membrane capacitance (left), peak depolarization and peak hyperpolarization in C2C12 myotubes subjected to 200 ms (right) of light stimulation in the absence (CTRL, grey) or loaded with 25 µM of Ziapin2 (red). Black dots represent the single values. Data are expressed as mean ± sd. Unpaired Student's *t* test/Mann–Whitney *U* test; ****$p < 0.0001$; $n = 17–18$ and 19–20 for CTRL and Ziapin2, respectively.

the electrolytic solution, with a diameter of 1 mm ($\Delta V_{applied} = 3$ V) and we performed the same analysis. We applied a pulsed square stimulation, each pulse was $20 \pm 0.1$ ms long, and we tested the three analysed frequencies. Figure 4 reports a schematic configuration of the setup used for the optical and the electrical stimulation and an example of both stimulations at 1 Hz. We observed a very high response efficiency at a stimulation frequency of 0.5 and 1 Hz, with mean contraction frequency of $0.49 \pm 0.03$ and $0.97 \pm 0.04$ Hz, respectively. Figure 3d shows that in both cases, more than 80% of the cells reached the target frequency of 0.5 and 1 Hz. At 2 Hz also the electrical stimulation loses efficiency, with a mean contraction frequency of $1.94 \pm 0.17$ Hz, still higher than in the optical experiment.

**Different contraction behaviour of the excited cells**. Due to the negligible spontaneous contraction activity of our cells, we can safely assign each contraction to the optical stimulation. However, the synchronization is far from perfect. Overall cell behaviour can be divided into six categories. The first group, A, is characterised by myotubes that perfectly follow the stimulus. In group B cells contract several seconds after the beginning of the stimulation. Group C is cells that stop contracting before the end of the stimulus. Group D collected erratic cells that starts with a delay and stop. Probably these last three classes of behaviour are related to the degree of maturation of the cells. If they are not enough differentiated, they cannot sustain the whole protocol. Other myotubes, group E, undergo a transitory freezing that lasts few seconds during the stimulation. Finally, group F collects myotubes with a random behaviour, apparently de-coupled from the driving force. The column bar graphs, Fig. 5, are the graphic representation of this distribution for both optical and electrical stimulation for all the studied frequencies. Furthermore, we reported a representative contraction trace for each case in

Supporting Fig. S3. Again, as previously noticed, the percentage of cells in case A drastically decrease applying a 2 Hz stimulation. This loss of stimulation efficiency is independent from the type of stimulation and therefore we attribute it to a low maturation level. The population of cells in case E is larger for optical stimulation than electrical one, for all stimulation frequency, and it is larger at higher frequency, going from 12.6% up to 29.3%. This can be ascribed to a default in the photo-isomerization, either because there are photophysical deactivating processes competing with photoisomerisation, or because back transfer from cis does not happen quickly enough. Even though some myotubes stop contracting for several seconds, when they start to contract again they keep following the light stimulation corroborating once more the deterministic effect of the photostimulation. Supporting Figure S4 reports a contraction trace of two myotubes showing a different contraction behaviour in the same field of view. The first myotube freezes for four seconds, whereas the second perfectly follows the light. It is clearly visible that even after the freezing period, the detected contractions resulted to be synchronized highlighting the correlation between light pulse and contraction.

**Energy evaluation and comparison of optical and electrical approach**. Finally, we compared the electrical and optical stimulation in terms of energy efficiency (i.e. required energy for triggering a single contraction), total energy absorbed during the stimulation and detrimental effect of the stimulation protocol.

The energetic aspects were investigated reverting to single pulse stimulation, either for optical or electrical pulses. We increased the applied energy rising the number of photons or the voltage. We ramped up progressively the energy and we checked if, in the field of view, a myotube starts to contract looking for the minimum energy required to trigger a contraction. Intuitively, we expect an increase in the probability of responding myotubes

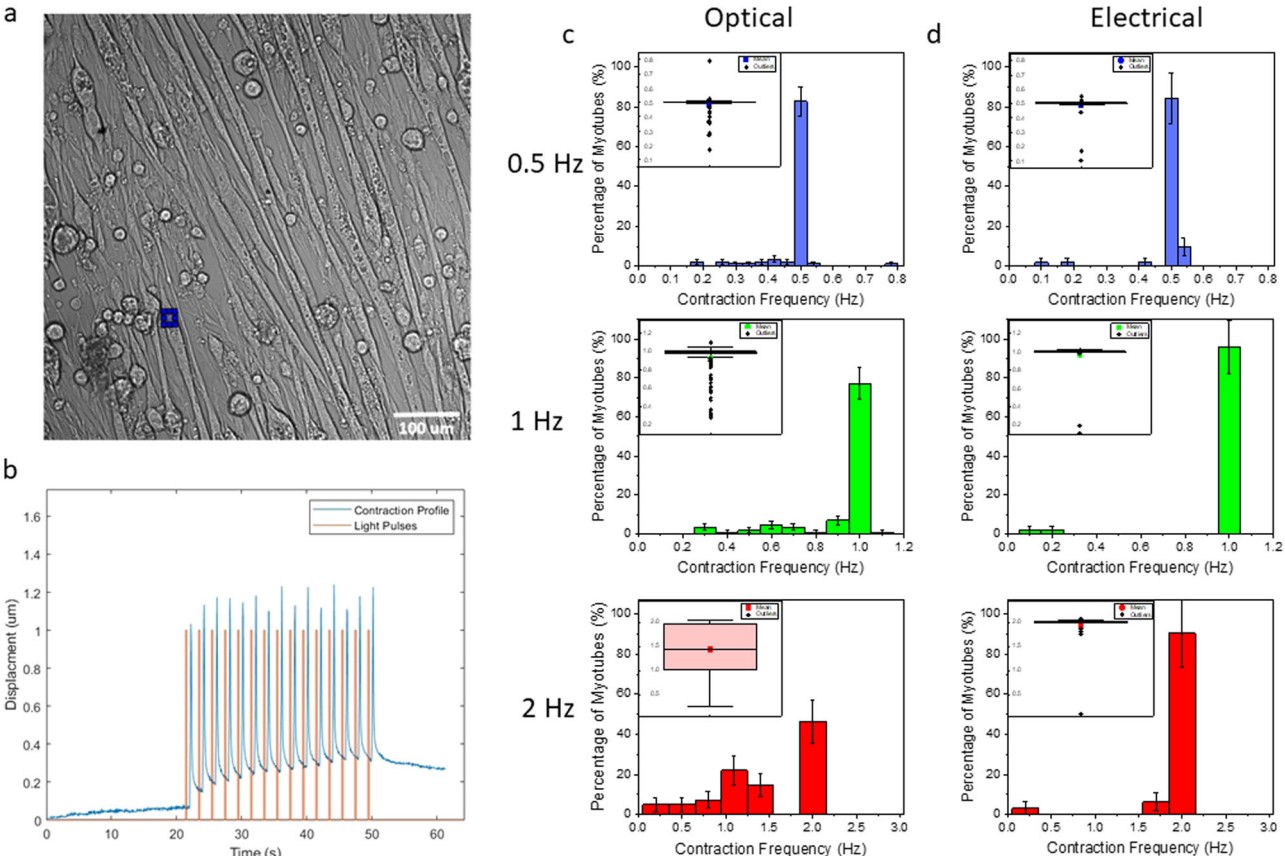

**Fig. 3 Contraction detection and analysis. a** Manual ROI selection on the contracting myotubes. It was used by Machine Learning code to follow the contraction of the cells in a video. **b** Contraction path (blue line) of the selected ROI during the stimulation and the stimulating protocol (red line). **c** The histograms show the mean contraction frequency of the myotubes stimulated with light at 0.5 1 and 2 Hz. (0.5 Hz $n = 87$ and $r = 5$, 1 Hz $n = 114$ and $r = 6$, 2 Hz $n = 41$ and $r = 4$). On the bottom left of each histogram there is the box plot rappresenting the single data. (square = mean, rhombous = outliers, range = 25/75) **d** The histograms show the mean contraction frequency of the myotubes stimulated with an electric field at 0.5, 1 and 2 Hz (0.5 Hz $n = 51$ and $r = 3$, 1 Hz $n = 50$ and $r = 3$, 2 Hz $n = 32$ and $r = 3$). On the bottom left of each histogram, there is the box plot representing the single data. (square = mean, rhombous = outliers, range = 25/75). The bin sizes are based on the standard error of the frequency, whereas the error on the bar is $(\sqrt{N})\%$.

upon increasing the applied energy. This is indeed confirmed, as seen in Fig. 6a, b, suggesting that cells have different thresholds to start contraction. A 100% activation probability means that, at that specific voltage or power density, we are always able to trigger at least one myotube activation. Further increase the applied energy resulted in a higher amount of responding cells without affecting the activation probability. In order to estimate the energy efficiency of the stimulation mechanisms we tried to evaluate the energy absorbed by the myotubes in the two situations.

To do this we need to know the average number of molecules internalized in the myotubes membrane. By measuring the concentration of Ziapin2 in the initial solution and the solution recovered after "washing" the sample we ended up with a rough estimate of $10^3 - 10^5$ molecules per cells. Knowing the applied light intensity, and the molecular absorption cross section[23], $\sigma = 2 \times 10^{-15}$ cm$^{-2}$, we can estimate the average energy absorbed by the myotubes. This value, under electrical stimulation, is estimated by working out the joules effect in the relevant volume. In order to work this out we made some geometrical considerations on the current path in the device, as reported schematically in Fig. 4b. We take into consideration the length of the wire of $2.4 \pm 0.1 \times 10^{-2}$ m and the distance between the wire of $7 \pm 2 \times 10^{-3}$ m. The cells are located $1 \pm 1 \times 10^{-3}$ m far from the wire and to simplify our assumption, we hypothesize that the electric field is constant and uniform in the region between the wire and

the myotubes sheets. By assuming a resistivity of the myotubes[57] film of $5.8 \pm 1.4 \times 10^{-1}$ $\Omega$ m, we can estimate the dissipated power, deeper explanation of the calculation can be found in the Supplementary Information. To sum up, to obtain 100% of cells activation, the estimated absorbed energy is $5.0 \pm 0.5 \times 10^{-6}$ J cm$^{-2}$ (optical) and $6.4 \pm 3.0 \times 10^{-6}$ J cm$^{-2}$ (electrical). The first value is related to a power density of $12.5 \pm 0.3$ mW mm$^{-2}$, the second to a voltage of $3.00 \pm 0.01$ V. Supplementary Fig. S5 reports the two energy trend line. This seems to indicate that optical excitation is more efficient, although the uncertainty in the estimate should probably lead to the conclusion that the two methods are comparable. It is indeed a reasonable result that regards the energy needed for the biological machine to cause contraction, regardless how this energy is delivered.

**Effects on the cells viability of the two stimulating methods.** A viability test has been conducted to understand if electrical and optical stimulation affect the cells condition in a different way. Furthermore, we are interested in evaluation which stimulation can be more appealing for a final application (i.e. robotics, drug testing and similar…). Since all these applications required a cells activity for period longer than the previously studied time interval (30 s), we decided to extent the stimulating protocol up to 20 min (0.5 Hz pulsed signal either for electrical and optical). We picked

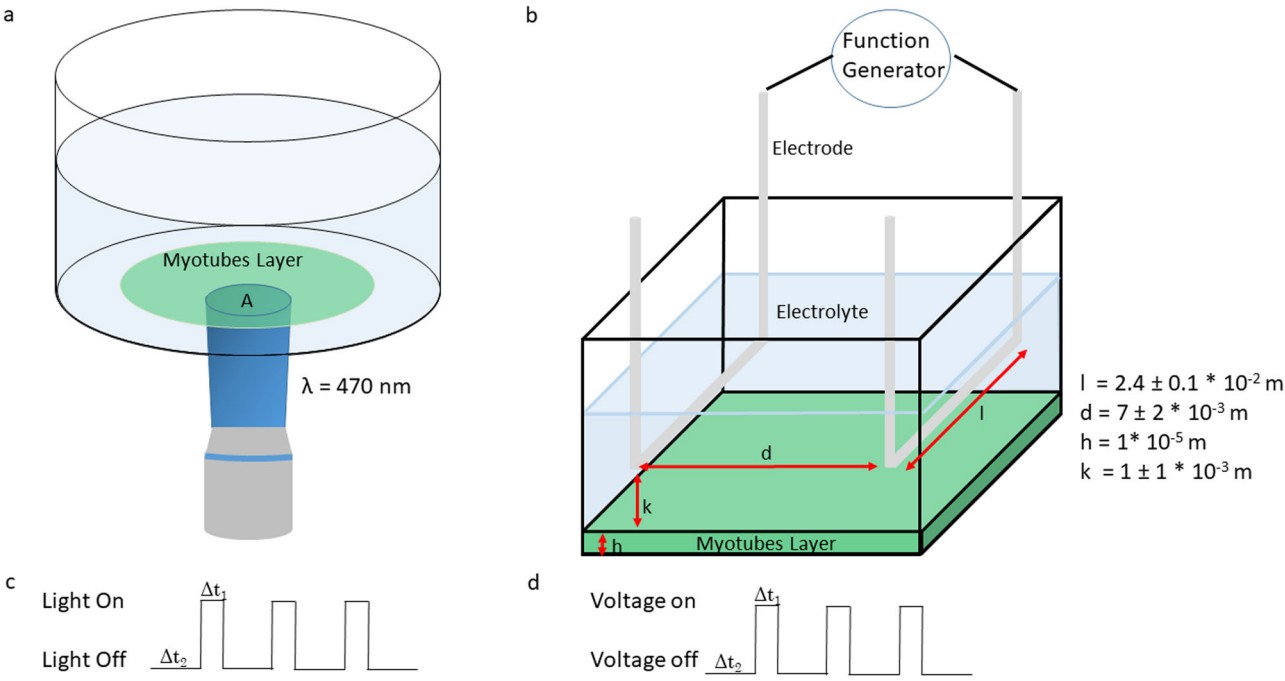

**Fig. 4 Set up configuration. a** Schematic view of the setup used to light cells' stimulation. A is the area of the LED spot (1.47 ± 0.03 mm² ×4 objective, 0.92 ± 0.02 mm² ×20 objective) and λ the wavelength used. **b** Schematic view of the setup used for the electrical stimulation. *k* is the distance between the electrode and the cells, *h* is the height of the cells layer, *l* is the horizontal length of the electrodes and *d* the distance between the electrodes. **c** Schematic representation of 3 light pulses at 1 Hz. $\Delta t_1 = 200$ ms, $\Delta t_2 = 800$ ms. **d** Schematic representation of 3 electric pulses at 1 Hz, $\Delta t_1 = 20$ ms, $\Delta t_2 = 980$ ms.

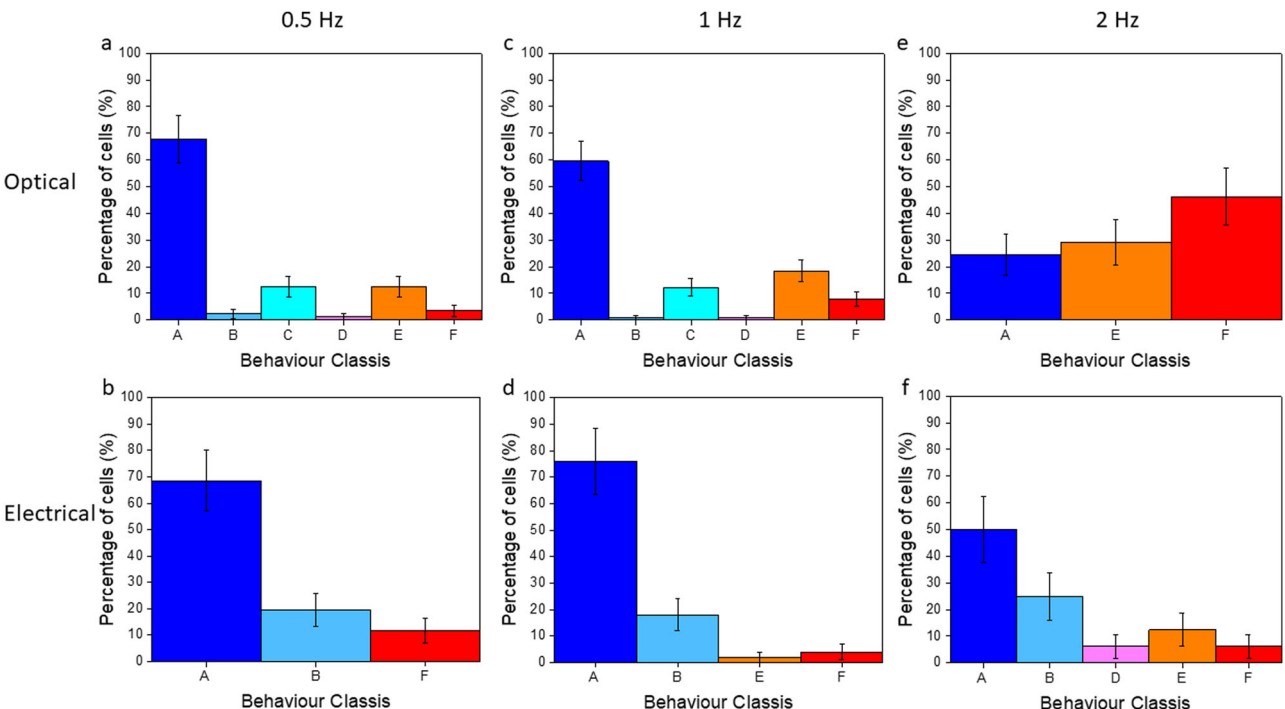

**Fig. 5 Representation of different types of myotubes contraction behaviour. a** 0.5 Hz optical stimulation ($n = 87$ and $r = 5$). **b** 0.5 Hz electrical stimulation ($n = 51$ and $r = 3$) **c** 1 Hz optical stimulation ($n = 114$ and $r = 6$). **d** 1 Hz electrical stimulation ($n = 50$ and $r = 3$) **e** 2 Hz optical stimulation ($n = 41$ and $r = 4$). **f** 2 Hz electrical stimulation ($n = 32$ and $r = 3$). The myotubes of case A perfectly follow the stimulation, the one in case B shows a delayed response to the stimulation. The cells in case C stop to contract before the end of the protocol, in case D the myotubes combine the behaviour of case B and C. The myotubes of case E stops to contract and then they start again during the stimulation. The myotubes of case F behave randomly. The error is reported as $\left(\sqrt{N}\right)\%$.

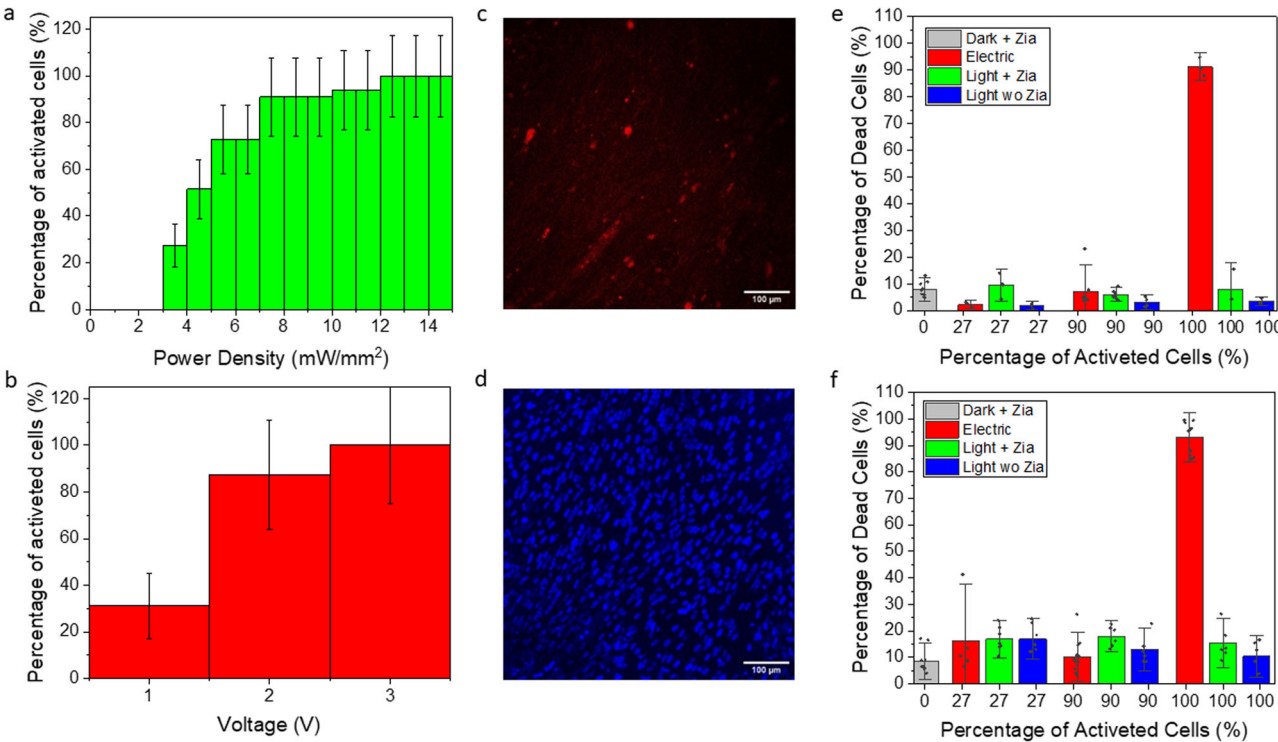

**Fig. 6 Cells activation and viability. a** Percentage of activated cells with different light power density ($f = 33$ and $r = 3$. The error as $(\sqrt{N})\%$. **b** Percentage of activated cells with different voltage ($f = 16$ and $r = 2$). The error as $(\sqrt{N})\%$. **c** Fluorescent image of dead cells' nuclei (stained with Propidium Iodide). **d** Fluorescent image of all the nuclei of a sample (stained with Hoechst). **e** Boxplot of the cell's death as a function of the stimulation type and activation percentage (Light 100% $f = 3$ and $r = 1$, 90% $f = 8$ and $r = 2$, 27% $f = 4$ and $r = 1$, 0% $f = 11$ and $r = 3$. Light without Ziapin2: Light 100% $f = 3$ and $r = 1$, 90% $f = 4$ and $r = 1$, 27% $f = 4$ $r = 1$. Electric 100% $f = 3$ and $r = 1$, 90% $f = 8$ and $r = 2$, 27% $f = 3$ ad $r = 1$, Dark with Ziapin $f = 9$ and $r = 3$) The whiskers are the sd, rhombus = single point, range 25/75. **f** Boxplot of the cell's death as a function of the stimulation type and activation percentage after 24 h the light/electrical stimulation (Light 100% $f = 6$ and $r = 2$, 90% $f = 6$ and $r = 2$, 27% $f = 7$ and $r = 2$, 0%. Light without Ziapin2: Light 100% $f = 7$ and $r = 2$, 90% $f = 6$ and $r = 2$, 27% $f = 6$ and $r = 2$. Electric 100% $f = 10$ and $r = 3$, 90% $f = 11$ and $r = 3$, 27% $f = 5$ and $r = 2$, Dark without Ziapin2 $f = 10$ and $r = 3$) The whiskers are the sd, rhombus = single point, range 25/75.

**Table 1 Comparison value between power density and voltage.**

| Percentage of activated cells (%) | Voltage (V) | Power density (mW/mm²) |
|---|---|---|
| 27 | 1 | 3.4 |
| 90 | 2 | 8.3 |
| 100 | 3 | 18.4 |

Voltage and power density needed to obtain the same percentage of cells activation.

up a low stimulation frequency because we have seen that the cells are more responsive at a lower frequency. Indeed, we aimed at analysing the effect of the two types of stimulation rather than the stress induced by a high frequency. The stimulated samples were stained with Hoechest 33342 dye, to visualize all cells nuclei, and with Propidium Iodide, to visualize the nuclei of the dead cells (Fig. 6c, d). We used the trackmate plugin[58], available for Fiji, to count the number of cells and evaluated the percentage of dead cells. The cells were stimulated at different light power and voltage to evaluate possible detrimental effects as a function of the energy release. The fraction of activated cells was used to compare the effect of the electric field and the light. Table 1 reports the correlations between percentage of contracting cells, voltage and power density.

We compare three different values: 27, 90 and 100% of activated cells and the results are reported in Fig. 6e. The

detrimental effect induced by Ziapin2 is not light dependent and it remains almost constant at a value of 10–15%. This value, even if it is not particularly low, is not significant compared to the control experiments, obtained keeping Ziapin2-treated cells in dark or shining light over not treated cells. We can reasonably associate this effect with the required steps for treating cells with Ziapin2. On the other hand, the toxicity of the electrical stimulation increases with the voltage up to 90% when the system is stimulated with 3 V.

Here we find the most relevant difference between electrical and optical excitation. Despite the energy release is almost comparable, the effect on the cells viability is quite different. The electrical stimulation has a maximum value of voltage that can be applied to the system, before biological damage sets in, probably related to electrolysis of the extracellular media, whereas the power of the light source can be increased without side effects, thus improving the responsivity of the molecules. 24 h after the cells' excitation, the vitality of the cells were tested another time. Indeed, several biological process may require time to affect cells. With the aim of avoiding further stress, the cells were placed in the incubator in a fresh medium after the stimulation. The nuclei were stained the day after the experiment, following the same procedure described previously. The results of our experiment are reported in Fig. 6f. The data confirms the harmful effect of the 3 V stimulation, while the toxicity level of the other cases is still low. The percentage of dead cells after 24 h increased compared to the same results obtained immediately after the stimulation, but it seems to be not dependent on the type of stimulation. We

believe that it is related to the stress employed on the cells i.e. switching several times the cells location and media.

**Free-standing devices as a first step through a functional device**. As a last test we have also add a Poly(*N*-iso-propylacrilamide) (PIPAAm) sacrificial layer underneath the PDMS substrate. The PIPAAm is a thermosensitive polymer showing hydrophobic behaviour if the temperature is higher than 32 °C and hydrophilic behaviour below 32 °C. This property candidates the polymer as a useful sacrificial layer in biological application where physiological temperature (37 °C) has to be kept for several days. The PIPAAm presence allows for the PDMS-myotubes double layer detachment leading to a free-standing biohybrid system[43]. We stimulated the device with a power density of 59.3 mW mm$^{-2}$, at 1 Hz, and we used a 4x objective to visualize a bigger area (Supplementary Video 4). We were able to obtain, as a proof of concept, a macroscopic movement of a freestanding portion of the device. In this case, we highlight that Ziapin2 can stimulate not only single myotubes but whole tissue, and it can be used to stimulate larger area and to control the actuation of a bigger device. Improving the maturation of the cells, it is possible to trigger a fully functional biohybrid robot with light thanks to the photochromic molecule.

## Discussion

We demonstrated that Ziapin2 is a promising tool for controlling and activating the contraction of skeletal muscle cells. It is a non-genetic approach, and it has advantages over opto-thermal stimulation, both via infrared heating of water or using nano-heaters, notably higher space and time resolution and limited phototoxicity. With Ziapin2 added to the cell culture, it is possible to obtain fairly high synchronization when the stimulation frequency is below 2 Hz. This limit however can be overcome by improving the differentiation process and myotubes maturation. The contraction-triggering mechanism associated to the generation of action potential has comparable requirements in terms of energy as the electrical one, suggesting that a natural threshold exists, in spite of the radically different physical phenomena involved. Ziapin2 act on the membrane capacitance to induce the resting voltage modulation, while the electrical current acts on the polarization of the cell membrane to evoke an action potential. Worth noticing by carrying out viability measurements that Ziapin2 phototoxicity is low, and at comparable performance it is less harmful than the electrical one. With light the intensity of the stimulus is enhanced without switching on cell degradation. For all these reasons, light can be an attractive alternative to electrical stimulation, particularly for the application of a 20 min pulses where the cells need to be stressed out without a breakdown.

This paper not only highlights the deterministic behaviour of C2C12 cells, mediated by light, but also demonstrates that light stimulation is preferable in terms of viability. These results clearly trigger Ziapin2 and, generally, photostimulation as a disruptive tool for all the applications requiring cells and tissue long-term stimulation and stability, like robotics, drug testing, energy conversion, and medicine. Indeed, if the contraction of the cells is fully regulated by the external stimulus, the actuation and the performance of the machinery are perfectly controlled. The freestanding device, described in the previous section, highlights the possible development of a biohybrid actuator based on skeletal muscle cells and triggered by light. Future perspective will employ light-triggered bio-hybrid systems for actuation (locomotion, swimming, gripping or grasping) as well as for biomedical applications, as self-propelled or light driven drug cargo and drug delivery systems. Moreover, they could be applied in environmental monitoring, in energy conversion as well as platform for developing artificial organs.

## Methods

**C2C12 cell culture and maturation**. C2C12 cells (ATCC) were cultured at 37 °C and 5% CO$_2$. Cells were kept under 70% of confluence to avoid damage and they were passed no more than 15 times. The maintenance medium contained Dulbecco's modified Eagle's medium (DMEM) supplemented with Fetal Bovine Serum (10%), glutamine (2%). The seeding density of the cells was 197 cell*mm$^{-2}$. When the cells reached the confluence, the medium was modified, into a starving medium, based on DMEM and glutamine (2%). After four days we switched to a differentiation media contained Dulbecco's modified Eagle's medium (DMEM) supplemented with Horse Serum (2%) and glutamine (2%). During the starving and differentiation process, the medium was changed every two days and it was 10 days long.

**Scaffold fabrication**. The scaffold was compound by a glass coverslip (Ø 18 mm), covered by a thin film of PDMS. Sylgard 184 polydimethylsiloxane was mixed at a ratio of 1:10 cross-linking agent and elastomer and it was spincoated on to the glass coverslip. The scaffold was baked in an oven for 4 h at 65 °C. It was sterilized under UV light for 1 h and, before the microcontact printing process, they were placed under UV-O for 8 min.

To obtain the free-standing sample we used 1 g of Poly(*N*-iso-propylacrilamide, PIPAAm) dispersed into 10 mL of 1-butanol. The solution was deposited by a spin coater on to the glass coverslip. Then the PDMS film was deposited with the same technique on top of it. The final steps were the same as the one stated above.

**Stamps fabrication**. The mold geometry for our microcontact printing process was performed with an ultrashort pulse laser system using the controlled ablation technique. PDMS was mixed in a 1:10 ratio between curing agent and elastomer and it was baked in a hoven at 65 °C for 4 h. Then the PDMS block was directly ablated by fs-micromachine.. The femtosecond laser facility was based on a generatively amplified Yb:KGW system (Pharos, Light Conversion, Vilnius, Lithuania) with 230-fs pulse duration, 515-nm wavelength (frequency doubled), and 100-kHz repetition rate focused with a 10x microscope objective (ML10X, Mitutoyo). Computer-controlled, 3-axis motion stages (ABL-1000, Aerotech, Pittsburgh, PA, USA) interfaced by CAD-based software (ScaBase, Altechna, Vilnius, Lithuania) with an integrated acousto-optic modulator was used to translate the sample relative to the laser irradiation desiderate patch. We worked in a constant density regime, impinging on the material a constant laser energy density (10,000 pulse/mm) as a function of the kinematics of the sample translocation. The average power and scan speed was of 200 mW and 10 mm/s respectively. We patterned the surface of the polymer creating a series of parallel line with average width of 84.7 ± 4.0 μm and lines distance of 27.8 ± 2.4 μm.

**Electrophysiology**. Standard patch clamp recordings were performed with an Axopatch 200B (Axon Instruments) coupled with a Nikon Eclipse Ti inverted microscope. Both C2C12 myoblasts and myotubes were measured in whole-cell configuration with freshly pulled glass pipettes (4–7MΩ), filled with the following intracellular solution [mM]: 12 KCl, 125 K-Gluconate, 1 MgCl2, 0.1 CaCl2, 10 EGTA, 10 HEPES, and 10 ATP-Na2. The extracellular solution contained [mM] 135 NaCl, 5.4 KCl, 5 HEPES, 10 Glucose, 1.8 CaCl2, and 1 MgCl2. The acquisition was performed with pClamp-10 software (Axon Instruments). Membrane

currents were low pass filtered at 2 kHz and digitized with a sampling rate of 10 kHz (Digidata 1440 A, Molecular Devices). A cyan LED coupled to the fluorescence port of the microscope and characterized by maximum emission wavelength of 474 nm provided the excitation light source. The illuminated spot on the sample has an area of 0.23mm$^2$ and a photoexcitation density of 50 mW/mm$^2$, as measured at the output of the microscope objective. Electrophysiological data are all expressed as mean ± sem. Normal distribution was assessed using the D'Agostino and Pearson's normality test. To compare two samples Student's *t* test or Mann–Whitney *U* test were used. The significance level was preset to $p < 0.05$ for all tests. Statistical analysis was carried out using GraphPad Prism 6 software.

**Optical and electrical stimulation**. The cells were treated with Ziapin2, diluted in water. We directly added it to the cells media reaching a final concentration of 25 μM. The multiwell was placed in the incubator for 10 min, time required for the internalization of the photochromic molecules, and then the cells were placed in the extracellular solution. The solution contained NaCl (0.14 M), KCl (0.0054 M), MgCl$_2$ (0.0018 M), CaCl$_2$ (0.0018 M), Glucose (0.011 M) and HEPES (0.01 M). Nikon Eclipse Ti inverted microscope, coupled with a cyan Led (470 nm) source, was used to stimulate and visualize the cells. We used a 20x/0.50 objective purchased from Nikon to visualize the cells' contraction, while the one used during the vitality test is a ×4 objective purchased from Nikon. We stimulated our myotubes at three different frequencies: 0.5, 1 and 2 Hz. 1 and 0.5 Hz. The light pulses were 200 ms long, whereas the pulses of the 2 Hz were 100 ms. The electrical stimulation was achieved applying a voltage of 3 V, and we tested 0.5, 1 and 2 Hz. Each electrical pulses were 20 ms long. The extracellular solution used during the electrical stimulation was the one previously reported. We record the effect of both stimulations thanks to PVcam software. We started the stimulation 20 s after the beginning of the video and we stopped the recording after several seconds at the end of the protocol, to understand if the contractions were generated by the external trigger or by some biological mechanism. All the measurements were performed at room temperature without $CO_2$ control.

**Video analysis**. The video was reconstructed through Fiji software and then it was analysed with a Matlab code. The first part of the code is based on a visual recognition algorithm that allows to follow the movement of an object, chosen by the user, through each frame. The code is based on Kanade–Lucas–Tomasi (KLT) algorithm[59–62]. The algorithm divides the selected ROI in subregions, and for each subregion evaluates its coordinates. The coordinates are then collected in a matrix that includes all the coordinates of each point of each frame. All the videos start from a resting position, we set it as a zero point of our system, and thanks to it we can evaluate the average displacement of each point from its zero. The average distance of the ROI is evaluated by the mean of the distance of each point. This strategy was used to obtain the average contraction path of contracting myotubes, Fig. 3b. In the supplementary section, Supporting Fig. S2.a, b, there is a picture of the ROI selected and the subregions used by the algorithm. The time between two consecutive contractions was obtained from the distance between two consecutive peaks from the contraction profile curve. To avoid the noise that can be collected from this analysis and to examine only the correct peaks, we implemented a series of quests necessary to highlight a peak (i.e. minimum value, minimum distance between two peaks, etc,). Then we evaluated the mean contraction time and frequency.

**Fluorescent analysis**. The fluorescent images have been taken by an inverted confocal laser microscope, Nikon Eclipse Ti2. The software used to acquire them is NIS-Element, Nikon Imaging Software. The objective used was a 20x. The fusion index was evaluated by counting the number of nuclei in each image. They were stained with Hoechst, and the cell's membrane were stained with CellMask Deep Red. The two dyes were incubated for 10 min in PBS and then washed away. The vitality test was performed by staining the nuclei with Hoechst and Propidium Iodide for 10 min. Both experiments were analysed with a Tracking plugin in Fiji to count the number of nuclei.

**Statistics and reproducibility**. Data are represented as mean ± standard error of the mean (s.e.m.) in the text as well as the electrophysiological data. The normal distribution of the electrophysiological data was assessed using the D'Agostino and Pearson's normality test. Two samples were compared by Student's *t* test or Mann–Whitney *U* test. In the histogram Fig. 3c, d, the error bar is given by $(\sqrt{N})\%$ where N is the number of data and reported as a percentage. The box plot presented in Fig. 3c, d highlights the mean value, the outliers and the range is 25–75%. In the column bar plot, Figs. 5 and 6a, b the error bar is given by $(\sqrt{N})\%$ where $N$ is the number of data and reported as a percentage. The boxplot, Fig. 6d, e the single value are reported by the rhombus and the whiskers indicate the sd, range 25/75. In the figures we used n as the number of cells analysed, *f* is the number of field studied and r the number of replica taken. We define a replica as a distinct seeding of the cells. The alignment has been evaluated on 10 fields and 1 replica, whereas the directionality of the random sample over 3 field and 1 replica. We used more than 30 sample for the frequency analysis and behaviour, and more than 3 distinct replica. The patch clamp analysis relies on more than 12 samples and 2 distinct replica. The viability test were conduct on more than 3 field and at least on 1 replica. The activation power analysis was conducted on at least 2 replica and more than 16 field. All the data can be found in the supplementary data file.

**Reporting summary**. Further information on research design is available in the Nature Portfolio Reporting Summary linked to this article.

## Data availability

The data that supports the findings of this study are available within the article, supplementary and supplementary data file. All the main raw data can be found at the following repository https://zenodo.org/record/8383422.

## Code availability

The code used for the frequencies analysis has been written in MatLab and can be found here https://zenodo.org/record/8383624.

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

## Acknowledgements

The authors thank Alessio Stomeo for the technical support during the polymerization of the PDMS of the stamps. G.L. and C.B. acknowledge the financial support of the PRIN "Membrane targeted light driven nanoactuators for neuro-stimulation" (Grant no. 2020XBFEMS).

## Author contributions

I.V. and V.V. contributed equally to this work. M.M. contributed to cells' growth, maintenance and patch clamp. I.V., S.B. and L.C. fabricated the PDMS stamps. S.P. took

the SEM images. C.B. designed and engineered the photochromic molecule. V.S. synthesized the Ziapin2. I.V. and V.V. wrote the paper. All the authors contributed to the revision and discussion. G.L. supervised and directed the work.

## Competing interests

The authors declare no competing interests.
