## [Peer Review File · Communications Biology]

Reviewers' comments:

Reviewer #1 (Remarks to the Author):

The authors show the use of Ziapin2 as a tool to allow the control of the contraction of C2C12 cells. They demonstrate that the cells contract upon shining light and follow the applied frequency. They also compare light stimulation to electrical stimulation to see which is safer and more efficient at contracting C2C12 cultures. Finally, they show that C2C12 treated with Ziapin2 can be used to actuate the PDMS arm, making this system interesting for biohybrid applications where actuation control can be done by simply shining light.

While control of cell activity with Ziapin2 is not a novel application, as the authors show in a previous study, the application for C2C12 is novel, showing the wide applicability of Ziapin2 for photostimulation of cells. Another novelty in this study that is quite interesting for all researchers studying the stimulation of cells is the comparison between light and electrical stimulation. The authors show that stimulation with Ziapin2 could be a better alternative to electrical stimulation, especially regarding viability issues. Thus, I recommend accepting this paper. However, the authors should address the comments below before this work can be published:

- The authors need to explain their electrical stimulation protocol clearer. They mention the use of Pt electrodes and later specify the dimensions in the energy comparison section. I suggest making a scheme with all the dimensions clearly marked. They also need to specify how high the electrodes are from the bottom of the cell culture well with ideally a photo/schematic. Additionally, the authors should show clearly what their electric pulses are. They mentioned 20ms pulses of 1-3V. I am assuming this is a square cathodic pulse. All these details are important for reproducibility.
- For the energy comparison, the authors compare energy successfully converted into stimulation for the cells. However, when discussing efficiency, it is more important to discuss the energy output of the stimulation device, i.e., power used by the electrodes or the LED. Thus, the authors need to show the power consumption of electrical stimulation at 3V for 20 minutes compared to 18.4mW/mm² LED usage for 20 minutes.
- The authors claim that light stimulation with Ziapin2-modified cells is safer for long-term stimulation. However, the longest period the authors tested was 20 minutes. This can not be considered long-term. The authors must show viability after at least 12 hours or change the claim to short-term stimulation.
- The authors claim that the mechanism of operation of Ziapin2 is the same as they mentioned in their previous paper, where they used neurons. However, it is not obvious that the mechanism with C2C12 is the same. Can the authors elaborate on why they believe the mechanism to be the same here or show evidence that shows that as they did in their previous paper?

Reviewer #2 (Remarks to the Author):

The authors describe a molecular phototransducer, used to stimulate C2C12 cells. The authors aimed to evaluate the effects of light intensity and stimulation frequency and they compared optical stimulation with an electrical one. The paper has merits, but it is also affected by important flaws, which prevent its possible publication, at this stage.

Major issues:

1) Most of the Introduction and in general the paper focus is on biohybrid devices and the possible advantages that molecular phototransducers could bring to their controllability. This is very true, indeed. However, the paper does not really contribute to this specific objective: no relevant contractions are achieved, and no clear comparison in terms of controllability is shown, over other strategies (e.g., optogenetic ones). Thus, the focus on biohybrid robotics is a bit overclaimed: it would be more appropriate to leave this possible application at the end of the paper, as a future perspective

(to be explored), rather than using it at the beginning, as the main claim, which is not supported by experimental data. Related to this point, the authors should re-formulate the novelty that they claim, which is basically the application of the molecular phototransducer (already known in the state-of-the-art) on a cell type on which it was not applied before. This of course reduces the novelty of this work.

2) The authors used a patterned PDMS substrate to guide myotube alignment and differentiation. Why the authors used channels with a width of $\sim 100 \mu\text{m}$? It is well known that smaller widths ($\sim 10 \mu\text{m}$) are much more efficient in guiding myotube alignment, and also in driving myotube maturation. Furthermore, it seems that fibronectin is not covalently attached to the PDMS surface. Did the authors evaluate if fibronectin detaches over time? (this may happen, due to PDMS hydrophobic recovery, and could further hamper myotube maturation). It would be nice to see an experiment in which a more appropriate substrate is used (this would also enable a more evident contraction).

3) The paper is too naïve from a molecular biology viewpoint: much more detailed analyses should be performed to assess cell differentiation (qRT-PCR analyses on the expression of genes crucial for myogenesis, quantification of myotube dimensions and fusion index, and possibly functionality tests – e.g., calcium imaging). As mentioned in the previous point, working on a different substrate would probably help in improving the differentiation level, but in any case, the differentiation should be more properly assessed. The authors mention that “There are possible actions to improve maturation, but this is out of scope here, so we focused on the optical triggering mechanism”. However, maturation cannot be neglected: many aspects analyzed by the authors (response to electrical/optical activation, metabolic activity, etc.) change much by changing the differentiation level.

Minor issues:

1) The article is well written, but there are some typos (e.g. miotubes instead of myotubes, death cells instead of dead cells, etc.). The authors should carefully check and fix them.

Reviewer #3 (Remarks to the Author):

This paper describes the methods to elicit contractions in muscle cells via utilizing azobenzene-based molecules, ziapin2. Since the membrane depolarization with ziapin2 has been already described in some of previous articles, this cannot be a novelty for this paper. The novelty I can find in here is the systematic comparisons of this optical ziapin2 stimulations with conventional electric stimulations. This is an incremental advancement, and the researchers who wants to utilize this method to their own experiments can be benefitted by referring this article. Still, there are some comments and recommendations, which may improve this paper further to resolve some questions and concerns.

1. Mechanisms of ziapin2 on how to elicit ionic influx into the cell were not explained at all.
2. Major advantages of optical stimulation are the spatio-temporal controllability. Authors showed the temporal control, and it would be good to show spatial control as well. The single-cellular level control would be interesting and beneficial to those want to try this method.
3. Lack of statistics in the data, especially in Figure 3. A pie chart is not a good method to show deviations and significance.
4. Additional cell viability tests on apoptosis are required. Propidium iodide is to investigate necrotic cells, only showing the terminal stages of dead cells. Some cells may undergo apoptosis as well, which may need to be properly addressed.
5. Many typos are found throughout the manuscript. Correct them both in main texts and supporting information.
6. Supplementary video 1 and 2 look the same. Please check.

Reply to Reviewer

We highlighted in blue the answers reviewer n.1, in red the answers to the reviewer n.2 and in green those to the third reviewer n.3.

Reviewer #1:

The authors show the use of Ziapin2 as a tool to allow the control of the contraction of C2C12 cells. They demonstrate that the cells contract upon shining light and follow the applied frequency. They also compare light stimulation to electrical stimulation to see which is safer and more efficient at contracting C2C12 cultures. Finally, they show that C2C12 treated with Ziapin2 can be used to actuate the PDMS arm, making this system interesting for biohybrid applications where actuation control can be done by simply shining light.

While control of cell activity with Ziapin2 is not a novel application, as the authors show in a previous study, the application for C2C12 is novel, showing the wide applicability of Ziapin2 for photostimulation of cells. Another novelty in this study that is quite interesting for all researchers studying the stimulation of cells is the comparison between light and electrical stimulation. The authors show that stimulation with Ziapin2 could be a better alternative to electrical stimulation, especially regarding viability issues. Thus, I recommend accepting this paper. However, the authors should address the comments below before this work can be published:

1. The authors need to explain their electrical stimulation protocol clearer. They mention the use of Pt electrodes and later specify the dimensions in the energy comparison section. I suggest making a scheme with all the dimensions clearly marked. They also need to specify how high the electrodes are from the bottom of the cell culture well with ideally a photo/schematic. Additionally, the authors should show clearly what their electric pulses are. They mentioned 20ms pulses of 1-3V. I am assuming this is a square cathodic pulse. All these details are important for reproducibility.

We thank the reviewer for this advice. We added in the text a sketch illustrating the set-up employed for optical and electrical stimulation (Figure 4 A and B), highlighting all the geometrical parameters in order to make them clearer. Moreover, we have adjusted the text accordingly (line 169-172 and line 239-244). In particular we explicitly mentioned that the electrical stimulation was performed placing both the electrodes in the conducting medium and applying a square pulse.

2. For the energy comparison, the authors compare energy successfully converted into stimulation for the cells. However, when discussing efficiency, it is more important to discuss the energy output of the stimulation device, i.e., power used by the electrodes or the LED. Thus, the authors need to show the power consumption of electrical stimulation at 3V for 20 minutes compared to 18.4mW/mm² LED usage for 20 minutes.

We acknowledge the reviewer's concern, and we agree that power consumption is a crucial consideration for the final application. To address this concern, we conducted direct measurements during our experiments by connecting an ammeter in series with the instruments used. The function generator consumed 0.12 A of current for electrical stimulation, while the LED consumed 0.08 A. It's important to note that the current requirements of the function generator can vary significantly depending on the specific device (namely the efficiency), the electronic circuit's geometry, and the materials employed. For this reason, we think that the information on the energy effectively

required to obtain the effect is the good figure of merit to compare the methods, while the final engineering of a device will have to take into account other parameters too.

3. The authors claim that light stimulation with Ziapin2-modified cells is safer for long-term stimulation. However, the longest period the authors tested was 20 minutes. This can not be considered long-term. The authors must show viability after at least 12 hours or change the claim to short-term stimulation.

We agree with this referee that a 12-hours stimulation could strengthen the presented results. However, a 12-hours (or longer) stimulation protocol will also affect the differentiation/maturation properties of the cells leading to results in which the overlap of the two effects could be hardly distinguished.

To completely avoid confusion and fulfil reviewer's request, we rephrase all the text (line 252-256 and line 327) related to the 20 minutes stimulation avoiding the "long-term" expression. We also test the viability of the cells, 24h after the stimulation to deeper investigate the possible toxic effects related to our experiments (line 291-299 and Figure 6F).

4. The authors claim that the mechanism of operation of Ziapin2 is the same as they mentioned in their previous paper, where they used neurons. However, it is not obvious that the mechanism with C2C12 is the same. Can the authors elaborate on why they believe the mechanism to be the same here or show evidence that shows that as they did in their previous paper?

We appreciate the reviewer for raising this concern, as it has prompted us to recognize the need for additional clarity in our manuscript. Our core assertion is that there is a common initial phenomenon across various cell models, which is the alteration in capacitance induced by mechanical perturbation of the membrane. We have substantiated this claim through capacitance measurements conducted on multiple cell types, including HEK cells, neurons, hPSC-CM, and various bacterial models. Additionally, we have performed patch clamp experiments on C2C12 myoblasts and myotubes, and the results, as outlined in the text (lines 116-118, 132-134, 376-389 and Figure 2), support our assertions.

It's essential to recognize that what occurs subsequently is indeed cell type-specific. In our work, we have primarily focused on the initial, common mechanosensitive response. The intricate details of the physiological processes that follow are beyond the scope of this study. We have reorganized and improved the presentation of these considerations in the text, as per your valuable advice (lines 50-51, 108-115).

Reviewer #2

The authors describe a molecular phototransducer, used to stimulate C2C12 cells. The authors aimed to evaluate the effects of light intensity and stimulation frequency and they compared optical stimulation with an electrical one. The paper has merits, but it is also affected by important flaws, which prevent its possible publication, at this stage.

Major issues:

1. Most of the Introduction and in general the paper focus is on biohybrid devices and the possible advantages that molecular phototransducers could bring to their controllability. This is very true, indeed. However, the paper does not really contribute to this specific objective: no relevant contractions are achieved, and no clear comparison in terms of controllability is shown, over other strategies (e.g., optogenetic ones). Thus, the focus on biohybrid robotics is a bit overclaimed: it would be more appropriate to leave this possible application at the end of the paper, as a future perspective (to be explored), rather than using it at the beginning, as the main claim, which is not supported by experimental data. Related to this point, the authors should re-formulate the novelty that they claim, which is basically the application of the molecular phototransducer (already known in the state-of-the-art) on a cell type on which it was not applied before. This of course reduces the novelty of this work.

We thank this referee for his/her suggestion for improving the quality of the paper. We carefully reorganize the structure of the paper focusing the introduction more on photostimulation (line 11-13, 21-32). Moreover we leave and implement the robotic perspective into the conclusion paragraph (line 329-339) as a future outlook of this work.

2. The authors used a patterned PDMS substrate to guide myotube alignment and differentiation. Why the authors used channels with a width of $\sim 100 \mu\text{m}$? It is well known that smaller widths ($\sim 10 \mu\text{m}$) are much more efficient in guiding myotube alignment, and also in driving myotube maturation. Furthermore, it seems that fibronectin is not covalently attached to the PDMS surface. Did the authors evaluate if fibronectin detaches over time? (this may happen, due to PDMS hydrophobic recovery, and could further hamper myotube maturation). It would be nice to see an experiment in which a more appropriate substrate is used (this would also enable a more evident contraction).

We thank the referee for the comment. We decided to use a line width of $100 \mu\text{m}$ following some published papers from Y. Sun et al.^{1,2}. The suggestion of the reviewer is however valuable for us. We agree that in literature several pattern dimensions have been reported to properly promote a good alignment and differentiation. For this reason, we added in the introduction a few lines (57-62) mentioning the other equally effective possible pattern dimensions and alignment strategy.

About the adhesion protein, we know from literature that other substrate and other adhesion protein (i.e. laminin) can be used to enhance the myotubes growth. However, since we based our work on already reported papers (PDMS and fibronectin successfully in literature^{3,4}) we focused mainly on just one substrate and adhesion protein. We further clarify this message in the text also adjusting the literature accordingly (line 86-87).

Anyway, we agree with this referee that the myotubes maturation improvement is a point that can be better investigated and we are actually working on optical methods for myotubes differentiation but we prefer to keep this results (still on-going) for a different work in order to keep the point of this work more simple and clear.

3. The paper is too naïve from a molecular biology viewpoint: much more detailed analyses should be performed to assess cell differentiation (qRT-PCR analyses on the expression of genes crucial for myogenesis, quantification of myotube dimensions and fusion index, and possibly functionality tests – e.g., calcium imaging). As mentioned in the previous point, working on a different substrate would probably help in improving the differentiation level, but in any case, the differentiation should be more properly assessed. The authors mention that “There are possible actions to improve maturation, but this is out of scope here, so we focused on the optical triggering mechanism”. However, maturation cannot be neglected: many aspects analysed by the authors (response to electrical/optical activation, metabolic activity, etc.) change much by changing the differentiation level.

We appreciate the referee for providing this comment. We agree with the reviewer that incorporating a more quantitative evaluation can enhance the overall quality of the paper. To address this concern, we have undertaken a characterization of myotube maturation by evaluating both the fusion index of our cells and the length of myotubes. The results of these analyses have been included in the revised text (lines 97-101), and a supporting figure (S1 B) has been added for further clarification.

Additionally, we have provided detailed information about the methods employed in the Material and Methods section, which now features a new subsection dedicated to this aspect (lines 425-431).

However, we respectfully disagree with the need for an extended investigation into cell maturation, as we believe it is secondary to the primary focus of our study – the opto-stimulation process. It's essential to note that our comparison between optical and electrical stimulation was conducted on cultures following the same management and differentiation protocol to ensure a meaningful comparison.

Minor issues:

1) The article is well written, but there are some typos (e.g. miotubes instead of myotubes, death cells instead of dead cells, etc.). The authors should carefully check and fix them.

We carefully check the entire text fixing the typos. Thank for noticing them.

Reviewer #3 (Remarks to the Author):

This paper describes the methods to elicit contractions in muscle cells via utilizing azobenzene-based molecules, ziapin2. Since the membrane depolarization with ziapin2 has been already described in some of previous articles, this cannot be a novelty for this paper. The novelty I can found in here is the systematic comparisons of this optical ziapin2 stimulations with conventional electric stimulations. This is an incremental advancement, and the researchers who wants to utilize this method to their own experiments can be benefitted by referring this article. Still, there are some comments and recommendations, which may improve this paper further to resolve some questions and concerns.

We thank the referee for acknowledging the usefulness of our work.

1. Mechanisms of ziapin2 on how to elicit ionic influx into the cell were not explained at all.

We add in the main text (line 44-49) a paragraph with a clear explanation of the Ziapin2-mediated photostimulation mechanism and related literature proving the assumptions.

2. Major advantages of optical stimulation are the spatio-temporal controllability. Authors showed the temporal control, and it would be good to show spatial control as well. The single-cellular level control would be interesting and beneficial to those want to try this method.

We appreciate the reviewer's suggestion to study the effect of patterned stimulation or single-cell excitation. However, the configuration of our current setup presents limitations that prevent us from selectively stimulating a small portion of the analyzed field of view. This limitation arises because both the stimulating wavelength and the bright field pass through the same optical path.

Nonetheless, we firmly believe that our claim remains reasonable, and the absence of experimental results in this specific context does not diminish the value of our study. It's important to note that the working mechanism of the effect we report supports our claims.

Furthermore, we would like to highlight that we have recently reported some evidence, albeit in a different cell model⁵, that supports the validity of our claims.

3. Lack of statistics in the data, especially in Figure 3. A pie chart is not a good method to show deviations and significance.

We thank the reviewer for the suggestion. We have converted the pie chart in a histogram (Figure 5), taking into account the statistical error. We also improve the statistic related to the calculation of the energy absorbed by the cells changing the error reported in Figure 6 accordingly.

4. Additional cell viability tests on apoptosis are required. Propidium iodide is to investigate necrotic cells, only showing the terminal stages of dead cells. Some cells may undergo apoptosis as well, which may need to be properly addressed.

We thank the reviewer for the suggestion. We evaluated possible delayed effect on cell viability analysing the amount of necrotic region 24h after the stimulation protocol. In particular, we take back the cells in the incubator after the stimulation and we performed the same viability test on these samples (cells stimulation + 24hours in the incubator). We believe that this protocol could assess this referee concerns about cells subjected to apoptosis process. The results highlighted that the presence of dead cells is still very low, it increases a bit from the previous study but homogeneously in all the conditions. Detail information regarding this experiment are reported in figure 6F and line 291-299.

5. Many typos are found throughout the manuscript. Correct them both in main texts and supporting information.

We carefully check the entire text fixing the typos. Thank for noticing them.

6. Supplementary video 1 and 2 look the same. Please check.

We thank the referee for noticing it. We probably had an error during the uploading process. We also better specify which video corresponds to which frequency (line 133). Furthermore we also add a caption describing all the videos in the supplementary information (line 53-57).

Bibliography

1. Sun, Y., Duffy, R., Lee, A. & Feinberg, A. W. Optimizing the structure and contractility of engineered skeletal muscle thin films. *Acta Biomater.* **9**, 7885–7894 (2013).
2. Choi, Y. S. *et al.* The alignment and fusion assembly of adipose-derived stem cells on mechanically patterned matrices. *Biomaterials* **33**, 6943–6951 (2012).
3. Feinberg, A. W. *et al.* Muscular Thin Films for Building Actuators and Powering Devices. *Science* **317**, 1366–1370 (2007).
4. Feinberg, A. W. *et al.* Controlling the contractile strength of engineered cardiac muscle by hierarchal tissue architecture. *Biomaterials* **33**, 5732–5741 (2012).
5. Vurro, V. *et al.* Light-triggered cardiac microphysiological model. *APL Bioeng.* **7**, 026108 (2023).

REVIEWERS' COMMENTS:

Reviewer #1 (Remarks to the Author):

The authors have taken my comments into account in the revised version.

Reviewer #2 (Remarks to the Author):

I am ok with the revisions made.

Reviewer #3 (Remarks to the Author):

The revision of the manuscript is well explained in terms of describing the mechanism of hyperpolarization/depolarization induction of ZIAPIN2 by demonstrating a patch clamp method following light stimulation. I don't have any objection for this paper to be published.

Reply to Reviewer

We highlighted in blue the answers reviewer n.1, in red the answers to the reviewer n.2 and in green those to the third reviewer n.3.

Reviewer #1:

The authors show the use of Ziapin2 as a tool to allow the control of the contraction of C2C12 cells. They demonstrate that the cells contract upon shining light and follow the applied frequency. They also compare light stimulation to electrical stimulation to see which is safer and more efficient at contracting C2C12 cultures. Finally, they show that C2C12 treated with Ziapin2 can be used to actuate the PDMS arm, making this system interesting for biohybrid applications where actuation control can be done by simply shining light.

While control of cell activity with Ziapin2 is not a novel application, as the authors show in a previous study, the application for C2C12 is novel, showing the wide applicability of Ziapin2 for photostimulation of cells. Another novelty in this study that is quite interesting for all researchers studying the stimulation of cells is the comparison between light and electrical stimulation. The authors show that stimulation with Ziapin2 could be a better alternative to electrical stimulation, especially regarding viability issues. Thus, I recommend accepting this paper. However, the authors should address the comments below before this work can be published:

1. The authors need to explain their electrical stimulation protocol clearer. They mention the use of Pt electrodes and later specify the dimensions in the energy comparison section. I suggest making a scheme with all the dimensions clearly marked. They also need to specify how high the electrodes are from the bottom of the cell culture well with ideally a photo/schematic. Additionally, the authors should show clearly what their electric pulses are. They mentioned 20ms pulses of 1-3V. I am assuming this is a square cathodic pulse. All these details are important for reproducibility.

We thank the reviewer for this advice. We added in the text a sketch illustrating the set-up employed for optical and electrical stimulation (Figure 4 A and B), highlighting all the geometrical parameters in order to make them clearer. Moreover, we have adjusted the text accordingly (line 169-172 and line 239-244). In particular we explicitly mentioned that the electrical stimulation was performed placing both the electrodes in the conducting medium and applying a square pulse.

2. For the energy comparison, the authors compare energy successfully converted into stimulation for the cells. However, when discussing efficiency, it is more important to discuss the energy output of the stimulation device, i.e., power used by the electrodes or the LED. Thus, the authors need to show the power consumption of electrical stimulation at 3V for 20 minutes compared to 18.4mW/mm² LED usage for 20 minutes.

We acknowledge the reviewer's concern, and we agree that power consumption is a crucial consideration for the final application. To address this concern, we conducted direct measurements during our experiments by connecting an ammeter in series with the instruments used. The function generator consumed 0.12 A of current for electrical stimulation, while the LED consumed 0.08 A. It's important to note that the current requirements of the function generator can vary significantly depending on the specific device (namely the efficiency), the electronic circuit's geometry, and the materials employed. For this reason, we think that the information on the energy effectively

required to obtain the effect is the good figure of merit to compare the methods, while the final engineering of a device will have to take into account other parameters too.

3. The authors claim that light stimulation with Ziapin2-modified cells is safer for long-term stimulation. However, the longest period the authors tested was 20 minutes. This can not be considered long-term. The authors must show viability after at least 12 hours or change the claim to short-term stimulation.

We agree with this referee that a 12-hours stimulation could strengthen the presented results. However, a 12-hours (or longer) stimulation protocol will also affect the differentiation/maturation properties of the cells leading to results in which the overlap of the two effects could be hardly distinguished.

To completely avoid confusion and fulfil reviewer's request, we rephrase all the text (line 252-256 and line 327) related to the 20 minutes stimulation avoiding the "long-term" expression. We also test the viability of the cells, 24h after the stimulation to deeper investigate the possible toxic effects related to our experiments (line 291-299 and Figure 6F).

4. The authors claim that the mechanism of operation of Ziapin2 is the same as they mentioned in their previous paper, where they used neurons. However, it is not obvious that the mechanism with C2C12 is the same. Can the authors elaborate on why they believe the mechanism to be the same here or show evidence that shows that as they did in their previous paper?

We appreciate the reviewer for raising this concern, as it has prompted us to recognize the need for additional clarity in our manuscript. Our core assertion is that there is a common initial phenomenon across various cell models, which is the alteration in capacitance induced by mechanical perturbation of the membrane. We have substantiated this claim through capacitance measurements conducted on multiple cell types, including HEK cells, neurons, hPSC-CM, and various bacterial models. Additionally, we have performed patch clamp experiments on C2C12 myoblasts and myotubes, and the results, as outlined in the text (lines 116-118, 132-134, 376-389 and Figure 2), support our assertions.

It's essential to recognize that what occurs subsequently is indeed cell type-specific. In our work, we have primarily focused on the initial, common mechanosensitive response. The intricate details of the physiological processes that follow are beyond the scope of this study. We have reorganized and improved the presentation of these considerations in the text, as per your valuable advice (lines 50-51, 108-115).

Reviewer #2

The authors describe a molecular phototransducer, used to stimulate C2C12 cells. The authors aimed to evaluate the effects of light intensity and stimulation frequency and they compared optical stimulation with an electrical one. The paper has merits, but it is also affected by important flaws, which prevent its possible publication, at this stage.

Major issues:

1. Most of the Introduction and in general the paper focus is on biohybrid devices and the possible advantages that molecular phototransducers could bring to their controllability. This is very true, indeed. However, the paper does not really contribute to this specific objective: no relevant contractions are achieved, and no clear comparison in terms of controllability is shown, over other strategies (e.g., optogenetic ones). Thus, the focus on biohybrid robotics is a bit overclaimed: it would be more appropriate to leave this possible application at the end of the paper, as a future perspective (to be explored), rather than using it at the beginning, as the main claim, which is not supported by experimental data. Related to this point, the authors should re-formulate the novelty that they claim, which is basically the application of the molecular phototransducer (already known in the state-of-the-art) on a cell type on which it was not applied before. This of course reduces the novelty of this work.

We thank this referee for his/her suggestion for improving the quality of the paper. We carefully reorganize the structure of the paper focusing the introduction more on photostimulation (line 11-13, 21-32). Moreover we leave and implement the robotic perspective into the conclusion paragraph (line 329-339) as a future outlook of this work.

2. The authors used a patterned PDMS substrate to guide myotube alignment and differentiation. Why the authors used channels with a width of $\sim 100 \mu\text{m}$? It is well known that smaller widths ($\sim 10 \mu\text{m}$) are much more efficient in guiding myotube alignment, and also in driving myotube maturation. Furthermore, it seems that fibronectin is not covalently attached to the PDMS surface. Did the authors evaluate if fibronectin detaches over time? (this may happen, due to PDMS hydrophobic recovery, and could further hamper myotube maturation). It would be nice to see an experiment in which a more appropriate substrate is used (this would also enable a more evident contraction).

We thank the referee for the comment. We decided to use a line width of $100 \mu\text{m}$ following some published papers from Y. Sun et al.^{1,2}. The suggestion of the reviewer is however valuable for us. We agree that in literature several pattern dimensions have been reported to properly promote a good alignment and differentiation. For this reason, we added in the introduction a few lines (57-62) mentioning the other equally effective possible pattern dimensions and alignment strategy.

About the adhesion protein, we know from literature that other substrate and other adhesion protein (i.e. laminin) can be used to enhance the myotubes growth. However, since we based our work on already reported papers (PDMS and fibronectin successfully in literature^{3,4}) we focused mainly on just one substrate and adhesion protein. We further clarify this message in the text also adjusting the literature accordingly (line 86-87).

Anyway, we agree with this referee that the myotubes maturation improvement is a point that can be better investigated and we are actually working on optical methods for myotubes differentiation but we prefer to keep this results (still on-going) for a different work in order to keep the point of this work more simple and clear.

3. The paper is too naïve from a molecular biology viewpoint: much more detailed analyses should be performed to assess cell differentiation (qRT-PCR analyses on the expression of genes crucial for myogenesis, quantification of myotube dimensions and fusion index, and possibly functionality tests – e.g., calcium imaging). As mentioned in the previous point, working on a different substrate would probably help in improving the differentiation level, but in any case, the differentiation should be more properly assessed. The authors mention that “There are possible actions to improve maturation, but this is out of scope here, so we focused on the optical triggering mechanism”. However, maturation cannot be neglected: many aspects analysed by the authors (response to electrical/optical activation, metabolic activity, etc.) change much by changing the differentiation level.

We appreciate the referee for providing this comment. We agree with the reviewer that incorporating a more quantitative evaluation can enhance the overall quality of the paper. To address this concern, we have undertaken a characterization of myotube maturation by evaluating both the fusion index of our cells and the length of myotubes. The results of these analyses have been included in the revised text (lines 97-101), and a supporting figure (S1 B) has been added for further clarification.

Additionally, we have provided detailed information about the methods employed in the Material and Methods section, which now features a new subsection dedicated to this aspect (lines 425-431).

However, we respectfully disagree with the need for an extended investigation into cell maturation, as we believe it is secondary to the primary focus of our study – the opto-stimulation process. It's essential to note that our comparison between optical and electrical stimulation was conducted on cultures following the same management and differentiation protocol to ensure a meaningful comparison.

Minor issues:

1) The article is well written, but there are some typos (e.g. miotubes instead of myotubes, death cells instead of dead cells, etc.). The authors should carefully check and fix them.

We carefully check the entire text fixing the typos. Thank for noticing them.

Reviewer #3 (Remarks to the Author):

This paper describes the methods to elicit contractions in muscle cells via utilizing azobenzene-based molecules, ziapin2. Since the membrane depolarization with ziapin2 has been already described in some of previous articles, this cannot be a novelty for this paper. The novelty I can found in here is the systematic comparisons of this optical ziapin2 stimulations with conventional electric stimulations. This is an incremental advancement, and the researchers who wants to utilize this method to their own experiments can be benefitted by referring this article. Still, there are some comments and recommendations, which may improve this paper further to resolve some questions and concerns.

We thank the referee for acknowledging the usefulness of our work.

1. Mechanisms of ziapin2 on how to elicit ionic influx into the cell were not explained at all.

We add in the main text (line 44-49) a paragraph with a clear explanation of the Ziapin2-mediated photostimulation mechanism and related literature proving the assumptions.

2. Major advantages of optical stimulation are the spatio-temporal controllability. Authors showed the temporal control, and it would be good to show spatial control as well. The single-cellular level control would be interesting and beneficial to those want to try this method.

We appreciate the reviewer's suggestion to study the effect of patterned stimulation or single-cell excitation. However, the configuration of our current setup presents limitations that prevent us from selectively stimulating a small portion of the analyzed field of view. This limitation arises because both the stimulating wavelength and the bright field pass through the same optical path.

Nonetheless, we firmly believe that our claim remains reasonable, and the absence of experimental results in this specific context does not diminish the value of our study. It's important to note that the working mechanism of the effect we report supports our claims.

Furthermore, we would like to highlight that we have recently reported some evidence, albeit in a different cell model⁵, that supports the validity of our claims.

3. Lack of statistics in the data, especially in Figure 3. A pie chart is not a good method to show deviations and significance.

We thank the reviewer for the suggestion. We have converted the pie chart in a histogram (Figure 5), taking into account the statistical error. We also improve the statistic related to the calculation of the energy absorbed by the cells changing the error reported in Figure 6 accordingly.

4. Additional cell viability tests on apoptosis are required. Propidium iodide is to investigate necrotic cells, only showing the terminal stages of dead cells. Some cells may undergo apoptosis as well, which may need to be properly addressed.

We thank the reviewer for the suggestion. We evaluated possible delayed effect on cell viability analysing the amount of necrotic region 24h after the stimulation protocol. In particular, we take back the cells in the incubator after the stimulation and we performed the same viability test on these samples (cells stimulation + 24hours in the incubator). We believe that this protocol could assess this referee concerns about cells subjected to apoptosis process. The results highlighted that the presence of dead cells is still very low, it increases a bit from the previous study but homogeneously in all the conditions. Detail information regarding this experiment are reported in figure 6F and line 291-299.

5. Many typos are found throughout the manuscript. Correct them both in main texts and supporting information.

We carefully check the entire text fixing the typos. Thank for noticing them.

6. Supplementary video 1 and 2 look the same. Please check.

We thank the referee for noticing it. We probably had an error during the uploading process. We also better specify which video corresponds to which frequency (line 133). Furthermore we also add a caption describing all the videos in the supplementary information (line 53-57).

Bibliography

1. Sun, Y., Duffy, R., Lee, A. & Feinberg, A. W. Optimizing the structure and contractility of engineered skeletal muscle thin films. *Acta Biomater.* **9**, 7885–7894 (2013).
2. Choi, Y. S. *et al.* The alignment and fusion assembly of adipose-derived stem cells on mechanically patterned matrices. *Biomaterials* **33**, 6943–6951 (2012).
3. Feinberg, A. W. *et al.* Muscular Thin Films for Building Actuators and Powering Devices. *Science* **317**, 1366–1370 (2007).
4. Feinberg, A. W. *et al.* Controlling the contractile strength of engineered cardiac muscle by hierarchal tissue architecture. *Biomaterials* **33**, 5732–5741 (2012).
5. Vurro, V. *et al.* Light-triggered cardiac microphysiological model. *APL Bioeng.* **7**, 026108 (2023).